# Mechanistic Insight into Permeation of Plasma-Generated Species from Vacuum into Water Bulk

**DOI:** 10.3390/ijms23116330

**Published:** 2022-06-06

**Authors:** Jamoliddin Razzokov, Sunnatullo Fazliev, Akbar Kodirov, Pankaj AttrI, Zhitong Chen, Masaharu Shiratani

**Affiliations:** 1Institute of Material Sciences, Academy of Sciences, Chingiz Aytmatov 2b, Tashkent 100084, Uzbekistan; akbarqodirov26@gmail.com; 2Institute of Fundamental and Applied Research, National Research University TIIAME, Kori Niyoziy 39, Tashkent 100000, Uzbekistan; 3Department of Physics, National University of Uzbekistan, Universitet 4, Tashkent 100174, Uzbekistan; 4College of Engineering, Akfa University, Milliy Bog Street 264, Tashkent 111221, Uzbekistan; 5Max Planck School Matter to Life, Jahnstrasse 29, 69120 Heidelberg, Germany; sunnatullo.fazliev@mtl.maxplanckschools.de; 6Faculty of Chemistry and Earth Sciences, Heidelberg University, Im Neuenheimer Feld 234, 69120 Heidelberg, Germany; 7Center of Plasma Nano-Interface Engineering, Kyushu University, Fukuoka 819-0395, Japan; chem.pankaj@gmail.com (P.A.); siratani@ed.kyushu-u.ac.jp (M.S.); 8Faculty of Information Science and Electrical Engineering, Kyushu University, Fukuoka 819-0395, Japan; 9Center for Advanced Therapy, National Innovation Center for Advanced Medical Devices, Shenzhen 518000, China; zt.chen@nmed.org.cn; 10Institute of Biomedical and Health Engineering, Shenzhen Institute of Advanced Technology, Chinese Academy of Sciences, Shenzhen 518055, China

**Keywords:** cold atmospheric plasma, plasma-generated reactive species, plasma-activated water, molecular dynamics, free energy profile

## Abstract

Due to their potential benefits, cold atmospheric plasmas (CAPs), as biotechnological tools, have been used for various purposes, especially in medical and agricultural applications. The main effect of CAP is associated with reactive oxygen and nitrogen species (RONS). In order to deliver these RONS to the target, direct or indirect treatment approaches have been employed. The indirect method is put into practice via plasma-activated water (PAW). Despite many studies being available in the field, the permeation mechanisms of RONS into water at the molecular level still remain elusive. Here, we performed molecular dynamics simulations to study the permeation of RONS from vacuum into the water interface and bulk. The calculated free energy profiles unravel the most favourable accumulation positions of RONS. Our results, therefore, provide fundamental insights into PAW and RONS chemistry to increase the efficiency of PAW in biological applications.

## 1. Introduction

Plasma, the fourth fundamental state of matter, consists of a complex mixture of high-energy particles (electrons, radicals, charged particles, excited atoms, molecules and photons). Cold atmospheric plasma (CAP) is a type of non-thermal plasma that comprises high-temperature electrons mixed with gaseous, high-energy chemicals of low temperature [1,2,3]. Although CAP is convenient for material processing, treating liquids, for example, water, with CAP produces a mixture of new reactive species. The resulting mixture is commonly called plasma-activated water (PAW) [4]. The interaction of CAP with water is a very complex physicochemical process resulting in UV radiation and numerous species with high chemical reactivity. Among the formed high-energy particles, reactive oxygen and nitrogen species (RONS) are particularly important, due to their wide variety of application areas, ranging from agriculture [5,6,7,8,9,10], to cancer [2,8,11,12,13,14,15], to the food industry [5,10,16]. Particularly in agriculture, it has been shown that PAW can be used to boost the stress tolerance [17], germination [18,19] and growth rate [20] of plants. In addition, there are a number of studies that demonstrate the positive synergy of PAW on the growth [10,19,21] and defence system of plants [5,6,9]. While the fertility and composition of soil deteriorate rapidly, PAW can offer a greener solution to the problems of agriculture and the food industry in a world with an ever-growing population. 

Although CAP itself has already shown promising results with cancer treatment, the application of PAW can really transform the cancer research field, since PAW is much more convenient to use, especially in the form of injections, but can still be as efficient as CAP treatment [22]. Many studies showed that PAW can kill various tumour types, such as ovarian [23], colon [24] and gastric cancer cells [25,26]. Recently, Kumar et al. showed that chemo-resistant pancreatic stellate cells can be killed by PAW [27]. These and many other successful cancer studies involving PAW might offer new therapeutic opportunities for cancer treatment, especially for tumours that cannot be beaten by conventional therapies. 

Other biological applications of PAW also rely on plasma-generated reactive species. In the food industry, for instance, PAW has been used for surface processing [16,28], meat treatment [29] and microbial inactivation [30,31,32]. Moreover, it has been shown that in many cases, the synergistic plasma effects (the reactions between the highly active species in PAW) are responsible for more pronounced antibiotic results compared to conventional disinfectants, such as H_2_O_2_, O_3_, etc. [33,34]

Over the past decades, plasma research has remarkably progressed and several techniques have been developed to create a highly reactive state of matter in mild conditions. Atmospheric pressure plasma jets [35], gliding arc [4] and dielectric barrier discharges [36] are amongst the most commonly used methods. Generally, the nature and composition of plasma depend on many parameters, such as feed gas, electrodes, exposure time, applied voltage, and so on. Hence, by changing these parameters, one can fine-tune the resulting mixture of reactive species and selectively make reactive oxygen species (ROS), reactive nitrogen species (RNS) or their mixtures. This is important to delineate how different plasma-generated species work in various applications, such as cancer treatment, microbial disinfection, and plant growth. For instance, Machala et al. showed a correlation with gaseous and aqueous RONS, plasma sources, and air flow conditions by testing the antibacterial effects of the generated PAW [37]. 

With regard to PAW, it has been shown that different CAP sources with similar gaseous RONS composition usually lead to PAWs with similar properties [22]. Hence, the way that plasma interacts with water is of the greatest significance for the properties of the resulting PAW. Therefore, during recent years, the plasma research community has become more focused on topics of interactions between the plasma and liquid interface, transport of plasma-generated gaseous species into water, and chemical reactions in the liquid phase. Studying these hot topics requires characterising the chemical composition of species in the plasma and liquid phases and having an understanding of chemical reactions that lead to the formation of RONS in the liquid phase. Usually, acid–base titrations, pH and electrical conductivity measurements are used to investigate the chemical nature of RONS in PAW. However, new instrumental techniques have been developed to characterise PAW. For example, Verlackt and co-workers developed a 2D axisymmetric fluid dynamics model to study the accumulation and chemical reactions of argon plasma jet-generated reactive species in a buffered aqueous solution, as well as transport from the gas phase [38]. Oh and collaborators developed the UV-Vis spectroscopy method to quantify the concentrations of the following major long-lived RONS in PAW generated by treatment with a helium plasma jet: H_2_O_2_, NO_2_^−^, NO_3_^−^, and O_2_ [39]. More recently, Oldham et al. presented an electrochemical technique to characterise electrochemical maps of an aqueous solution in contact with an atmospheric pressure plasma jet [40]. This method allows the spatial distribution of redox reactions taking place in such plasma–liquid interfaces to be identified. 

Delineating the transportation mechanisms of plasma-generated species into bulk water is equally important. Knowledge about the permeation of plasma-generated species is essential to understand the storage of PAW, which has many implications, particularly in medicine, where, for instance, PAW is used as injections. The permeation processes take place at the vacuum–water interface and are difficult to characterise with common analytical techniques. Molecular dynamics (MD) is a powerful tool to understand, simulate and model such nanoscale processes. MD simulations can pave the way for experiments and provide insights into underlying permeation mechanisms that are responsible for the formation of RONS in PAW. 

In this study, we aim to investigate the permeation of plasma-generated RONS from vacuum into the water bulk through the calculation of free energy profiles by performing MD simulations. 

## 2. Simulation Setup

We carried out MD simulations, in order to study the transport capability of the following RONS from vacuum into the water bulk: H_2_O_2_, HO_2_, OH, O_3_, NO, NO_2_, *trans*-HNO_2_, *cis*-HNO_2_, HNO_3_, *trans-perp*-ONOOH, *cis-perp*-ONOOH (throughout the main text, *tp* for *trans-perp* and *cp* for *cis-perp* notations are used) and N_2_O_4_. GROMACS program package (GPU version) [41] was employed to perform MD simulations by introducing GROMOS united atom force field parameters for RONS developed in the literature [42,43,44,45]. Initially, the cubic box was filled with an SPC water model [46] by making use of Packmol software [47]. Further, we performed energy minimization and a short 100 ps equilibration run by applying NVT and NPT ensembles. Next, the 50 ns production run proceeded, employing the velocity rescaling thermostat [48] at 300 K and the Parrinello–Rahman barostat [49] at 1 atm pressure, respectively. The equilibrated model system was transferred into the centre of the tetragonal box with the dimensions x = 3.2, y = 3.2, and z = 8.0 nm, to create the vacuum and the water interface (see Figure 1). The current model system was equilibrated again for 50 ns to ensure the adequate distribution of water molecules between the vacuum and water bulk. It is a standard approach to study the behaviour of molecules between the vacuum, water interface and bulk [50,51]. Moreover, it is quite complex to mimic a real condition in computer simulations due to the large number of particles in model systems. However, the parameters of the current model system are sufficient to study the permeation process of molecules into the liquid content [43].

The tetragonal simulation box is used to perform umbrella sampling (US) simulations [52]. The current simulation technique allows us to calculate the free energy profile (FEP) along the reaction coordinate. The centre of mass of water molecules and the centre of mass of RONS were used as reaction coordinates in our US simulations. The position restrained potential with a force constant of 2000 kJ mol^−1^ nm^−2^ was applied to the reaction coordinates along the z-axis. Thus, RONS move on the x-y plane while the motion of the latter is limited to the z-axis. In order to use the computational resources efficiently and collect more data in simulations, we inserted one of the seven RONS separated by a 1 nm distance along the z-axis to the model system (see Figure 1a). Before using the model system for US simulations, the subsequent energy minimization and short NVT simulation were carried out. Next, the system was run for 6 ns and the last 4 ns trajectory was used for data analysis. The US simulations were repeated 12 to 20 times by randomizing positions of RONS on the x-y plane (cf Figure 1a,b). The location of RONS varies between −3.5 and 3.5 nm along the z-axis, which spans from the vacuum into the water phase, ending up in the vacuum again by shifting the position of RONS by 0.05 nm. Thus, the individual FEP was obtained using 7 × 12 = 84 US windows. The final FEP was built by averaging 12 (i.e., H_2_O_2_, HO_2_˙, OH˙, *trans*-HNO_2_, *cis*-HNO_2_, HNO_3_, *tp*-ONOOH, *cp*-ONOOH and N_2_O_4_) to 20 (i.e., O_3_, NO and NO_2_) individual FEPs by employing the weighted histogram analysis method (WHAM). Overall, we performed 0.864 and 1.44 μs of 1800 US simulations for the abovementioned RONS. A visual molecular dynamics tool was used to prepare images of the model system [53].

## 3. Results and Discussion

It must be mentioned that in experimental conditions, the chemical reactions of RONS might occur in the gaseous and the liquid phases, as well as at the gas–liquid interface. However, these reactions cannot take place in classical MD simulations, which do not consider quantum-level chemical potentials. Despite these limitations, US simulations could assist in obtaining the RONS permeation rate from the vacuum into the water bulk, through the calculations of FEPs. According to the FEP data, one can predict the potential accumulation positions of RONS in the model system.

As is clear from Figure 2, the FEPs of all the investigated RONS decrease at the vacuum–water interface, showing the minimum values of ΔG, which facilitates the accumulation of RONS. This enhancement of the solute molecule concentration at the vacuum–water interface is a general trend and is applicable to all the RONS studied. The concentration of O_3_, NO and NO_2_ at the vacuum–water interface is higher than that of the gaseous phase, but with corresponding ∆G_gs_ values of (free energy change for transition from the gaseous phase to the water surface, i.e., vacuum–water interface) −3.94, −1.21 and −4.66 kJ/ mol, respectively. In contrast, the concentrations of the other species at the vacuum–water interface are ~10^3^–10^6^ times higher compared to the concentration in the gaseous phase, with ∆G_gs_ values being on the order of 10 kJ/mol (Table 1). It is, therefore, obvious that the vacuum–water interface is the most favourable accumulation site for all the RONS studied here.

Based on the permeabilities of the water bulk, we divided the studied RONS into two groups: (i) hydrophilic species with ∆G_hydr_ < 0 (hydration free energy, which corresponds to the transport of a compound from the gaseous phase into the solvent, i.e., water in this case) and (ii) hydrophobic species with ∆G_hydr_ > 0. We try to discuss our results by following this grouping.

The FEPs of the hydrophilic RONS, i.e., OH˙, HO_2_˙, H_2_O_2_, *trans*-HNO_2_, *cis*-HNO_2_, HNO_3_, *tp-ONOOH*, *cp-ONOOH* and N_2_O_4_, follow the general trend by decreasing at the water interface where FEPs have minimum values. Moreover, according to the FEPs, a shift in the free energy minimum from the water surface towards the water bulk phase (cf. Figure 2, Figure 3 and Figure 4 below) can be observed. They show different hydration free energies depending on their chemical composition and polarity.

The FEP profiles of three closely related hydrophilic species—OH˙, HO_2_˙ and H_2_O_2_—become more negative in the given order (Figure 2). We speculated that the presence of an extra oxygen atom in HO_2_˙ and H_2_O_2_ causes stronger dispersion interactions and hydrogen bonding ability [42] than OH radicals in water. Further deeper penetration of these ROS into the water bulk is hindered due to the varying free energy barriers from the vacuum–water interface into the liquid phase (∆G_sl_) (see Figure 2a). This barrier is in the order of 5.53 and 5.76 kJ/ mol for OH and HO_2_ radicals, respectively, while H_2_O_2_ has ∆G_sl_ of only 0.45 kJ/ mol. The hydration of H_2_O_2_ is the strongest among all the RONS studied here: ∆G_hydr_ = −36.24 kJ/ mol (which very well reproduces −36.5 kJ/ mol that is calculated from the experimentally determined Henry’s law constant reported by Sander [54]). This might be an underlying mechanism where there is only a little energy barrier for H_2_O_2_ to enter the water bulk from the vacuum–water interface. Such a large ∆G_hydr_ and small ∆G_sl_ of H_2_O_2_ resulted in a concentration enhancement of 2 × 10^6^ and 1.7 × 10^6^ at the vacuum–water interface and in the water bulk, respectively, compared to the gaseous phase (Table 1).

Amongst hydrophilic RNS, hydration free energies show dependence on the polarity of the species (Figure 3). For instance, N_2_O_4_—the least polar RNS here—also has a larger ∆G_hydr_ compared to the other RNS. The trend is still in place in the case of nitrogen acids. It is interesting to note that peroxinitrous and nitric acids demonstrate different ∆G_hydr_ as well as ∆G_sl_, despite having the same empiric formula—HNO_3_. ∆G_sl_ of peroxinitrous acid is large, ~8 kJ/ mol, which is very close to that of hydrophobic species (see Table 1), but its ∆G_hydr_ is still favourable enough, ~−18 kJ/ mol; whereas, nitric acid has the second most favourable ∆G_hydr_ of −26.87 kJ/ mol and the second smallest ∆G_sl_ of 5.29 kJ/ mol. We invoked basic concepts of chemical structure and electronics to account for the observed differences in these compounds. In water, if deprotonated, peroxinitrous acid with a peroxide bond results in a not very stable anion. Thus, it spends most of the time as peroxinitrous acid, whose hydration is favourable, but not as high as nitric acid. This occurs because the latter, being a strong acid, deprotonates easily in water and forms an anion that is significantly stabilised by resonance. The structure of peroxinitrous acid is similar to that of nitrous acid, thus their FEP profiles are similar and close to each other.

It is remarkable how basic concepts of chemical bonding can be applied to explain subtle differences in the behaviours of *cis-trans* isomer forms of nitrous acid. In *trans*-HNO_2_, individual bond dipoles are aligned for a favourable interaction with each other. Such favourably aligned bond dipoles result in a pronounced molecular dipole, which ultimately leads to favourable hydration. However, the *cis* form produces an unfavourable alignment of bond dipoles, thereby making it less polar than the *trans* form. Our simple qualitative explanation correlates well with experimentally determined dipole moments: 1.85 D [56] and 1.42 D [57] for *trans* and *cis* isomers, respectively. Moreover, the differences in chemical structures of these isomers also contribute to the different topological polar surfaces of isomers, with the *trans* form having a larger polar surface than the *cis* form. These electronic and structural differences are displayed in the ∆G_hydr_ and ∆G_sl_ of these isomers: *trans* isomer has more favourable hydration (∆G_hydr_ = −16 kJ/ mol) and smaller ∆G_sl_ of 7.09 kJ/ mol, while *cis* isomer has less favourable hydration (∆G_hydr_ = −11.7 kJ/ mol) and experiences a larger ∆G_sl_ barrier of 7.24 kJ/ mol.

It is interesting to note that the simulations of *cis-trans* nitrous acid forms show marked differences in FEPs, while those of *cp* and *tp* forms of peroxinitrous acid resemble each other. The structures of *cp* and *tp* conformers of peroxinitrous acid are very similar, where appropriate bond orientations allow the molecule to possess appreciable dipole moment. Furthermore, one can expect them to yield dipole moments of similar value. Using higher levels of wave-function-based electronic structure theory and density functional theory, the calculated dipole moments of *cp*-ONOOH and *tp*-ONOOH were found to be 1.64 D and 1.71 D, respectively, which are very close to each other [58]. Similarities in the structure and electronics of these two conformers are reflected in their similar FEP profiles. Although *tp*-ONOOH demonstrates slightly better concentration enhancement at the water surface and bulk, this might be within the error of calculation, since the values for the two conformers are very close (Table 1).

Figure 4 shows the different natures of hydration of the hydrophobic species: O_3_, NO and NO_2_. These species can only enter the vacuum–water interface, exhibiting minimum free energy in FEPs. However, the ∆G_sl_ values are much larger: 7.92, 8.68 and 7.27 kJ/ mol for O_3_, NO and NO_2_, respectively. This results in overall positive ∆G_hydr_ values, which directly correlate with small solubilities of O_3_, NO and NO_2_ in water, and thereby impede their transport from the vacuum into the water bulk (Figure 4 and Table 1). These hydrophobic RONS can only accumulate at the interface between the vacuum and water. Overall, our findings correlate very well with the experimental data, but with small deviations. The only substantial difference is the ∆G_hydr_ of HNO_3,_ which differs from the experimental data by ~10 kJ/mol. Currently, we do not have a convincing explanation for this discrepancy.

The chemical behaviour of the RONS in water bulk also influences their hydration. It is apparent that the chemical reactions take place in PAW. As a result, other types of RNS, such as NO_2_^−^, NO_3_^−^, N_2_O_3_ and ONOO^−^, might form in PAW [10]. Some RONS, due to their low permeation profile, cannot permeate into water bulk themselves, but they might act as precursors for the other RONS that are dominant in water bulk. We refrain from discussing the very complex chemistry of PAW, which is nicely reviewed elsewhere [2,8,10,12,59,60].

Our results correlate with the experimental findings of Oinuma et al. [61], where quantitative analysis of the plasma–water interaction shows that the H_2_O_2_ is predominantly produced through the recombination of OH radicals either in the gaseous or in the liquid phase, and is one of the major ROS in PAW. Taking FEP of H_2_O_2_ (which gives thermodynamic information) and relative reactivities of RONS into account, we can speculate that at larger spatial plasma–water interactions, i.e., at longer plasma jet distances, H_2_O_2_ can become the most dominant species in PAW. Although we do not have kinetic data, at moderate exposure times (in the order of minutes), equilibrium should be quickly established and we expect the RONS composition in PAW to correlate with their FEPs. This prediction is nicely demonstrated in a recent preprint by Lamichhane et al., where they show that the ratio of NO_2_^−^ dominates at shorter plasma jets, while the H_2_O_2_ concentration becomes much higher at longer plasma jets [62]. Trey Oldham and Elijah Thimsen also reported similar results and suggested that H_2_O_2_ forms not close to the atmospheric pressure plasma jet centre line, but rather in the region surrounding the plasma–liquid interface [40]. Regarding exposure time, Oh et al. also experimentally showed that exposure time affects the total RONS concentration in PAW, but not the composition of RONS in PAW [39], which implies the formation of a rapid equilibrium.

CAP devices operate in open air environments, producing a cocktail of RONS, UV and electric fields. Thus, it is complicated to define the composition of the RONS generated in PAW. The selective production of RONS would increase the effects of RONS in practice and provide tailor-made applications. For instance, NO, NO_2_, NO_3_˙, NO_3_^−^ and ONOO^−^ showed immunogenic cell death in cancer cells and were also beneficial in increasing crops on the field [63,64]. On the other hand, OH˙, O_3_ and H_2_O_2_ species play an essential role in bacterial decontamination and wound healing purposes [65,66,67]. Hence, our investigation results assist in developing an efficient approach to the selective production of ROS and RNS.

It is obvious that some of the considered RONS form after penetration into the water phase. Our main purpose is to study the most favourable place of RONS accumulation in PAW. This knowledge can assist in designing PAW production experiments that generate desired RONS more efficiently. For example, O_3_, NO and NO_2_ mostly accumulate at the vacuum–water interface and do not penetrate into the water bulk. Therefore, when producing these molecules, it might be more efficient to use water droplets as a PAW medium. This would facilitate O_3_, NO and NO_2_ formation since the water interaction surface is increased and the molecules will concentrate on the thin layer of water. Meanwhile, to make PAW with hydrophilic RONS (OH˙, HO_2_˙, H_2_O_2_, HNO_3_, etc.), one can already use plasma jets with water bulk. These findings coupled with different plasma–water interaction parameters, such as exposure time and gas-type jet–water distance, can offer controlled PAW production and allow us to fine-tune RONS composition.

Further development of experiments to obtain PAW can benefit from computational studies on plasma–sample interactions and how various parameters affect the outcome. For instance, Vichiansan et al., using finite element analysis and statistics, showed that plasma feed gas velocity and plasma gap affect RONS production [68]. In particular, NO production increases linearly as the distance from the nozzle tip to the ground increases, while that of OH radical and H_2_O_2_ reaches a maximum production density at around 15 mm plasma gap [68]. Qian et al. also numerically studied OH radical production, but considered the H_2_O content in pulsed direct current atmospheric pressure plasma jets (APPJ), and showed that the density of OH radicals gradually increases as the H_2_O content increases [69]. Recently, molecular beam mass spectrometry was applied to characterise ion compositions in the APPJ with a varying feed gas content: Ar+O_2_, Ar+H_2_O and Ar+air [70]. This study showed that ROS ions such as O^−^, O_3_^−^ and O_2_^+^ could be observed in the close-to-substrate region, which is consistent with the other numerical and experimental studies [40,71]. There are also other simulation studies that report the gas flow rate insignificantly influencing discharge ignition and propagation, at least for short plasma gaps [72]. Therefore, we suggest that by modulating various parameters, selective ROS and RNS production can be achieved; in particular, shorter plasma jets, shorter exposure time, more N_2_ content in the plasma feed gas, and shorter distances between the plasma and the sample result in higher density of RNS, while more ROS production requires more H_2_O and O_2_ content in the plasma feed gas, longer exposure time, and larger distances between the plasma and the sample.

## 4. Conclusions

We performed MD US simulations in order to reveal the permeation of CAP-generated individual RONS from the vacuum into the water bulk. According to the obtained FEPs data, we can conclude that hydrophilic RONS, i.e., OH, HO_2_, H_2_O_2_, *trans*-HNO_2_, *cis*-HNO_2_, HNO_3_, ONOOH and N_2_O_4_, are able to enter to the vacuum–water interface and the water bulk phase. The considerably low free energy values at the vacuum–water interface show that this is the most favourable site for the accumulation of these RONS. The translocation probabilities of the other RONS, i.e., O_3_, NO and NO_2_ into the water bulk are comparatively less favourable due to the free energy barrier.

Using concepts of chemical bonding and electronics, we put forward plausible explanations for different hydration and surface-to-liquid transport properties of the RONS. Based on our results, we also provided suggestions to design experiments for PAW production with the controlled composition of RONS.

Our findings correlate well with the existing experimentally determined free energy values of some RONS and can, therefore, serve as starting points for the other RONS that still lack experimental data. The current study plays an important role in the development of specific types of RONS-enriched PAW that can be applied in plasma medicine and plasma agriculture fields.

## Figures and Tables

**Figure 1 ijms-23-06330-f001:**
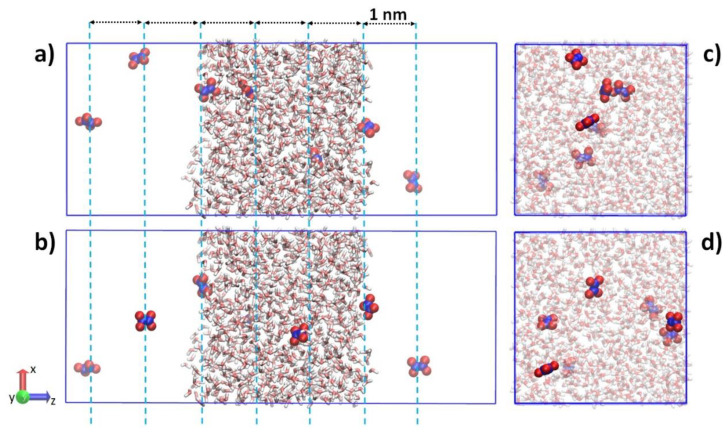
(**a**) The positions of seven N_2_O_4_ inserted into the model system and separated by a 1 nm distance along the z-axis. (**b**) The same positions were used to insert seven N_2_O_4_ along the z-axis, but randomized on the x-y plane. (**c**,**d**) Top views of the model system. N_2_O_4_ is represented in VDW view and water molecules are illustrated in licorice view.

**Figure 2 ijms-23-06330-f002:**
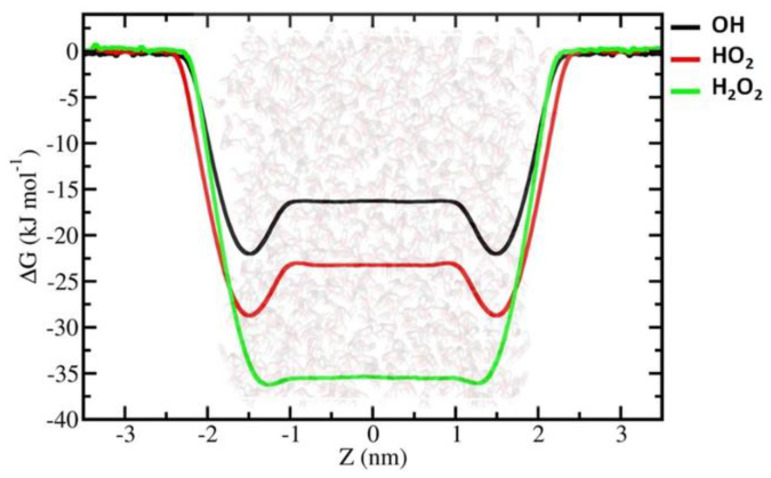
FEPs of the hydrophilic ROS obtained by US simulations by calculation of the WHAM method.

**Figure 3 ijms-23-06330-f003:**
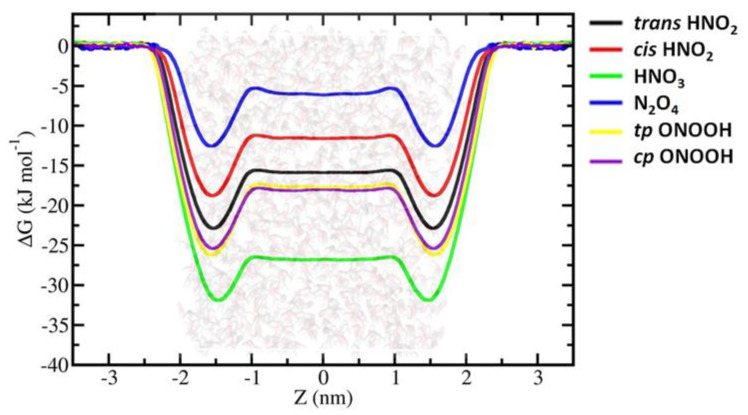
FEPs of the hydrophilic RNS obtained by US simulations by calculation of the WHAM method.

**Figure 4 ijms-23-06330-f004:**
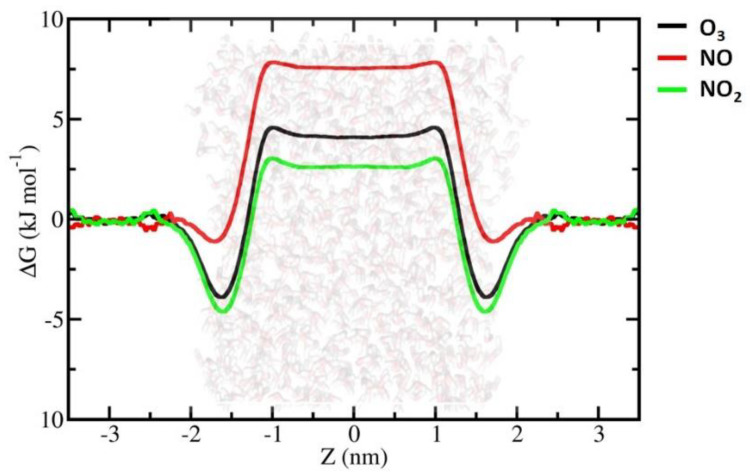
FEPs of the hydrophobic RONS were obtained by US simulations by calculation of the WHAM method.

**Table 1 ijms-23-06330-t001:** Free energy p values with ∆G_hydr_ were calculated from experimental Henry’s law constants and concentration enhancement of species at the vacuum–water interface and in the water bulk.

Species	∆G_gs_, kJ/ mol	∆G_sl_, kJ/ mol	∆G_hydr_, kJ/ mol	∆G_hydr(expt)_ *, kJ/ mol	Concentration Enhancement **	
					Surface	Bulk
O_3_	−3.94	7.92	3.98	3.46	4.8	0.20
NO	−1.21	8.68	7.47	7.61	1.6	0.05
NO_2_	−4.66	7.27	2.61	3.01	6.4	0.35
OH ˙	−22.04	5.53	−16.51	−17.09	6880.9	749.5
HO_2_˙	−28.97	5.76	−23.21	−24.29	110,740.7	10,999.3
H_2_O_2_	−36.21	0.45	−35.76	−36.50	2,018,128.0	1,684,976.9
N_2_O_4_	−12.70	6.70	−6.00	−8.86	162.7	11.1
*trans*-HNO_2_	−23.06	7.09	−15.97	−18.0 ***	10,357.3	603.6
*cis*-HNO_2_	−18.93	7.24	−11.69	−11.2 ***	1977.5	108.5
HNO_3_	−32.16	5.29	−26.87	−36.42	397,883.2	47,714.6
*tp-ONOOH*	−26.55	8.05	−18.50	Not found	41,969.4	1664.4
*cp-ONOOH*	−25.47	7.45	−18.02	Not found	27,219.6	1373

* These ∆G_hydr_ values are calculated from experimental Henry’s law constants from the Compilation of Henry’s law constants (version 4.0) by Sander [54] if not stated otherwise. ** Concentration enhancement represents the ratio of the concentration of species at the water surface and water bulk to that of the gaseous phase. *** Data from [43,55].

## Data Availability

Not applicable.

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
