# Peer review of "Mechanistic Insight into Permeation of Plasma-Generated Species from Vacuum into Water Bulk"

_ijms, 2022, doi:10.3390/ijms23116330_

Round 1
Reviewer 1 Report
1.the title is improper. It is suggested to change as “The mechanistic insight on diffusion of species generated in plasma from gas into the water bulk”
2.it is suggested to change the concept “cold atmospheric plasmas” as “ cold atmospheric pressure plasmas”
3.p 2 line 28, the research is stated at atmospheric pressure, then how to explore “ the permeation of RONS from vacuum into …”. Vacuum and atmospheric pressure is different, not only discharge mode, but also the composition of species.
Besides, it is difficult to study the permeation of species from vacuum into the water bulk. The saturation vapor pressure of water is high, ~24 Torr, in the vacuum condition water will be evaporated rapidly.
If in vacuum, what is the pressure in the chamber?
4.Actually some of RONS are formed in water bulk after species diffusion from gas into water, not form in the gas and then permeate into the water.
5.p1 line36, the definition of plasma in this sentence “Plasma…consists of a complex mixture of high energy particles (radicals, charged particles, excited atoms, and molecules)….” is suggested to change as “Plasma…consists of a complex mixture of high energy particles (electrons, charged particles, radicals, excited atoms and molecules, photons)….”
6.p2 line 80, this sentence is argued “….different CAP sources with similar gaseous RONS composition, …lead to PAWs of similar properties…..” In fact the composition generated in plasma is greatly dependant on the plasma source.
7.P4 line156,“…the chemical reactions of RONS might occur on the water interface or in the water bulk” is imprecise, RONS can also occur in the gaseous phase
8.the word of “permeation capability” might be improper
9.Besides the larger free energy ΔGhydr of O3, NO and NO2 which causes relatively less favourable in water bulk as said, the solubility shall be considered. As known the solubility of O3, NO not NO2 are relatively small.
10.what the force field parameters used in MD shall be given.
11.the permeation rate of species such as O3, NO and NO2 did not show, while it is concentration in the text.
Author Response
|
Dr. Jamoliddin Razzokov Academy of Sciences Chingiz Aytmatov 2b, 100084 Tashkent, Uzbekistan jrazzokov@gmail.com tel. +998901162320 |
|
|
|
DATE 20 May 2022 |
SUBJECT Manuscript revision |
|
Dear Editor,
We would like to thank you and both referees for the evaluation of our manuscript submitted to International Journal of Molecular Sciences (Manuscript ID: ijms-1727631) by Jamoliddin Razzokov, Sunnatullo Fazliev, Akbar Kodirov, Pankaj Attri, Zhitong Chen and Masaharu Shiratani, entitled “The mechanistic insight on permeation of plasma generated species from vacuum into the water bulk”.
We read both referees’ reports carefully. The referees recommended our paper for publication after considering some comments. Please find below our answers to the referees’ comment, as well as the corresponding changes and improvements made in the manuscript. Moreover, we are grateful to the anonymous referees for their constructive comments which helped improving the quality of our manuscript.
Sincerely,
Razzokov Jamoliddin
Reply to the Referees
We would like to thank referee for the useful comments/suggestions that, indeed, are valuable and improve the quality of our manuscript. We have taken all comments into consideration and revised the manuscript accordingly. All additions and modifications to the manuscript are formatted in red text.
Reviewer 1:
1.the title is improper. It is suggested to change as “The mechanistic insight on diffusion of species generated in plasma from gas into the water bulk”
Reply: We thank the referee for the recommendation. We did not directly calculate diffusion coefficient of reactive oxygen and nitrogen species (RONS). Here, we concentrated on calculation of free energy profiles to determine potential barrier to get more detailed information.
2.it is suggested to change the concept “cold atmospheric plasmas” as “ cold atmospheric pressure plasmas”
Reply: Actually, the term of cold atmospheric plasma was started using by scientists (Alexander Friedman, Michael Keidar and Mounir Laroussi) who established the “Plasma medicine” research field. There is a special conference called “International conference on Plasma Medicine” which is organized biennially. The scientists and board members of this conference encouraged others to use the term “Cold atmospheric plasma - CAP” or “Non-thermal plasma - NTP” in their plenary talks. These both standard terms facilitate finding papers related on “Plasma medicine” and “Plasma agriculture” field. Therefore, we hope the referee understands if we leave it as it is.
3.p 2 line 28, the research is stated at atmospheric pressure, then how to explore “ the permeation of RONS from vacuum into …”. Vacuum and atmospheric pressure is different, not only discharge mode, but also the composition of species.
Besides, it is difficult to study the permeation of species from vacuum into the water bulk. The saturation vapor pressure of water is high, ~24 Torr, in the vacuum condition water will be evaporated rapidly.
If in vacuum, what is the pressure in the chamber?
Reply: It is quite complex to mimic a real condition in computer simulations due to the large number of particles in the model system. Therefore, we initially equilibrated the water bulk in NPT ensemble. Then, in order to create a vacuum and the water interface we expanded equilibrated system with fixing a box size. It is a standard approach to study behaviors of molecules between vacuum, water interface and bulk (see Angewandte Chemie International Edition, 56(50), pp.15846-15851; J. Phys. Chem. A 2004, 108, 11573-11579). Moreover, the density of water remains as it is which represents real atmospheric conditions. To make this more clear we added explanation to the main text, see page :
It is a standard approach to study behavior of molecules between vacuum, water interface and bulk [50, 51]. Moreover, it is quite complex to mimic a real condition in computer simulations due to the large number of particles in model systems. However, the parameters of the current model system is sufficient study permeation process of molecules into the liquid content [43].
4.Actually some of RONS are formed in water bulk after species diffusion from gas into water, not form in the gas and then permeate into the water.
Reply: Indeed, the referee is right. Some RONS might form in the interface of water after the reaction with other RONS. Our point is to define the potential accumulation site of those RONS. We added the following sentences by mentioning it in the main text, see page 9:
It is obvious that some of the considered RONS form after the penetration into the water phase. Our main purpose is to study the most favorable place of RONS accumulation in PAW.
5.p1 line36, the definition of plasma in this sentence “Plasma…consists of a complex mixture of high energy particles (radicals, charged particles, excited atoms, and molecules)….” is suggested to change as “Plasma…consists of a complex mixture of high energy particles (electrons, charged particles, radicals, excited atoms and molecules, photons)….”
Reply: We thank the referee for this suggestion. The text has been changed to include additional particles, see page 1:
Plasma, the fourth fundamental state of matter consists of a complex mixture of high energy particles (electrons, radicals, charged particles, excited atoms, molecules and photons)
6.p2 line 80, this sentence is argued “….different CAP sources with similar gaseous RONS composition, …lead to PAWs of similar properties…..” In fact the composition generated in plasma is greatly dependant on the plasma source.
Reply: We understand that the referee is stressing the fact that the chemical composition of RONS depends on many parameters, including the plasma source. Actually, we provided this information in the previous paragraph (no. 5). However, we agree with the referee that the opening clause of this sentence was misleading. In this sentence what we wanted to deliver to the reader was: plasmas with similar chemical composition generates plasma activated water with similar properties. Therefore, we changed the sentence to following to make it clearer, see page 2:
With regard to PAW, it has been shown that different CAP sources with similar gaseous RONS composition, usually, lead to PAWs of similar properties.
7.P4 line156,“…the chemical reactions of RONS might occur on the water interface or in the water bulk” is imprecise, RONS can also occur in the gaseous phase
Reply: We thank the referee for this comment. We changed the sentence as following, see page 4:
It must be mentioned that in experimental conditions, the chemical reactions of RONS can occur in the gaseous and liquid phases, as well as on the gas-liquid interface.
8.the word of “permeation capability” might be improper
Reply: The referee is right. We changed that combination with “permeation” in two instances, see page 3 and 10:
In this study, we aim to investigate the permeation of plasma-generated RONS from vacuum into the water bulk through the calculation of free energy profiles by performing MD simulations.
and
We performed MD US simulations in order to reveal the permeation of CAP generated individual RONS from vacuum into the water bulk.
9.Besides the larger free energy ΔGhydr of O3, NO and NO2 which causes relatively less favourable in water bulk as said, the solubility shall be considered. As known the solubility of O3, NO not NO2 are relatively small.
Reply: We thank the referee for the comment. Since there is a direct correlation between ΔGhydr and solubility, we refrained ourselves from restating it. But we have modified the sentence according to the comment to make it clearer, see page 8:
This results in overall positive ∆Ghydr values, which directly correlate with small solubilities of O3, NO and NO2 in water, and thereby impede their transport from vacuum into the water bulk (Fig 4 and Table 1)
10.what the force field parameters used in MD shall be given.
Reply: Indeed, we did not mention about the force field parameters. Now, it is included in the main text, see page 3.
…. MD simulations by introducing GROMOS united atom force field parameters for RONS developed in literatures …
11.the permeation rate of species such as O3, NO and NO2 did not show, while it is concentration in the text.
Reply: We assume the referee is commenting on the 2nd paragraph of the Results and Discussion section. It is true that we did not perform kinetics studies here and we stated it in the main text. And we did not report the concentration of any RONS in this paper. We could not clearly identify where exactly the referee was commenting, but with high probability we assumed it to be this sentence: “In contrast, the concentration of the other species at the vacuum-water interface is ~103-106 times higher compared to the concentration in the gaseous phase with ∆Ggs values of on the order of 10 (Table 1).” If it is so, then we would like to stress that here we are describing the difference between concentrations in various phases. These, so called “concentration enhancements” are calculated based on ∆G values considering equilibrium (the definition is given in the footnote of the Table 1). We did not write an actual concentration. We hope this clarifies the misunderstanding.

Reviewer 2 Report
Dear Authors,
The manuscript ijms-1727631-peer-review-v1, entitled: ‘The mechanistic insight on permeation of plasma generated species from vacuum into the water bulk’ provide fundamental insights on PAW and RONS chemistry to increase the efficiency of PAW in biological applications. The authors performed molecular dynamic simulations in order to study the permeation of reactive oxygen and nitrogen plasma species (RONS) from vacuum to water and bulk medium. By determining the free energy profiles, they could reveal the most favorable conditions of RONS accumulation.
The main text of the manuscript is well written, the model is well presented, and the simulation condition well stated. The figures in the manuscript are of good quality and provides info on the stated model. The influence of the plasma on the production of RONS is described and discussed, in comparison to the literature.
The conclusions are consistent and in concordance with the results of the proposed model.
I would, however, expect more results to be presented.
The manuscript can be considered for publication in the IJMS journal after adding more insight on this model. More results on the geometry of the plasma, the interface between plasma source and the sample to be treated. What about the working gas? Add mixture of N2, O, OH?.
In the present form, the ijms-1727631-peer-review-v1 cannot be considered for publication.
After the above-mentioned issue addressed, the contribution could be considered:
MINOR REVISION.
Author Response
|
Dr. Jamoliddin Razzokov Academy of Sciences Chingiz Aytmatov 2b, 100084 Tashkent, Uzbekistan jrazzokov@gmail.com tel. +998901162320 |
|
|
|
DATE 20 May 2022 |
SUBJECT Manuscript revision |
|
Dear Editor,
We would like to thank you and both referees for the evaluation of our manuscript submitted to International Journal of Molecular Sciences (Manuscript ID: ijms-1727631) by Jamoliddin Razzokov, Sunnatullo Fazliev, Akbar Kodirov, Pankaj Attri, Zhitong Chen and Masaharu Shiratani, entitled “The mechanistic insight on permeation of plasma generated species from vacuum into the water bulk”.
We read both referees’ reports carefully. The referees recommended our paper for publication after considering some comments. Please find below our answers to the referees’ comment, as well as the corresponding changes and improvements made in the manuscript. Moreover, we are grateful to the anonymous referees for their constructive comments which helped improving the quality of our manuscript.
Sincerely,
Razzokov Jamoliddin
Reply to the Referees
We would like to thank referee for the useful comments/suggestions that, indeed, are valuable and improve the quality of our manuscript. We have taken all comments into consideration and revised the manuscript accordingly. All additions and modifications to the manuscript are formatted in red text.
Reviewer 2:
Dear Authors,
The manuscript ijms-1727631-peer-review-v1, entitled: ‘The mechanistic insight on permeation of plasma generated species from vacuum into the water bulk’ provide fundamental insights on PAW and RONS chemistry to increase the efficiency of PAW in biological applications. The authors performed molecular dynamic simulations in order to study the permeation of reactive oxygen and nitrogen plasma species (RONS) from vacuum to water and bulk medium. By determining the free energy profiles, they could reveal the most favorable conditions of RONS accumulation.
The main text of the manuscript is well written, the model is well presented, and the simulation condition well stated. The figures in the manuscript are of good quality and provides info on the stated model. The influence of the plasma on the production of RONS is described and discussed, in comparison to the literature.
The conclusions are consistent and in concordance with the results of the proposed model.
I would, however, expect more results to be presented.
The manuscript can be considered for publication in the IJMS journal after adding more insight on this model. More results on the geometry of the plasma, the interface between plasma source and the sample to be treated. What about the working gas? Add mixture of N2, O, OH?.
In the present form, the ijms-1727631-peer-review-v1 cannot be considered for publication.
After the above-mentioned issue addressed, the contribution could be considered:
MINOR REVISION.
Reply: We thank the referee for their valuable suggestions. In the discussion section, we have provided information how plasma feed gas, exposure time, distance between the plasma and the sample (in our case it is water) affect the properties of PAW and what kind of RONS composition can be obtained. However, we agree with the referee and updated our discussion section with one more paragraph dedicated to the geometry of the plasma, the interface between plasma source and the sample, and the working gas to make our discussion more thorough. The new paragraph is given here, see page 9:
Further development of experiments to obtain PAW can benefit from computational studies on plasma-sample interactions and how various parameters affect the outcome. For instance, Vichiansan et al., using finite element analysis and statistics showed plasma feed gas velocity and plasma gap affect RONS production (Vichiansan, Leksakul et al. 2021). In particular, NO production increases linearly at the distance from the nozzle tip to the ground increases, while that of OH radical and H2O2 reaches a maximum production density at around 15 mm plasma gap (Vichiansan, Leksakul et al. 2021). Qian et al., also numerically studied OH radical production, but considering H2O content in a pulsed direct current atmospheric pressure plasma jets (APPJ) and showed that the density OH radicals gradually increases as H2O content increases (Qian, Yang et al. 2016). Recently, molecular beam mass spectrometry was applied to characterize ion compositions in the APPJ with varying feed gas content: Ar+O2, Ar+H2O and Ar+air (Jiang, Aranda Gonzalvo et al. 2021). This study showed that ROS ions like O−, O3−, O2+ could be observed at the close-to-substrate region which is consistent with the other numerical and experimental studies(Kelly and Turner 2014, Oldham and Thimsen 2022). There are, also, other simulation studies which report gas flow rate have insignificantly influences discharge ignition and propagation, at least for short plasma gaps (Yan and Economou 2017). Therefore, we suggest that by modulating various parameters selective ROS and RNS production can be achieved, particularly, shorter plasma jets, shorter exposure time, more N2 content in the plasma feed gas and shorter distances between the plasma and the sample result in higher density of RNS, while more ROS production requires more H2O and O2 content in the plasma feed gas, longer exposure time, larger distances between the plasma and the sample.
Jiang, J., Y. Aranda Gonzalvo and P. J. Bruggeman (2021). "Analysis of the Ion Conversion Mechanisms in the Effluent of Atmospheric Pressure Plasma Jets in Ar with Admixtures of O2, H2O and Air." Plasma Chemistry and Plasma Processing 41(6): 1569-1594.
Kelly, S. and M. M. Turner (2014). "Generation of reactive species by an atmospheric pressure plasma jet." Plasma Sources Science and Technology 23(6).
Oldham, T. and E. Thimsen (2022). "Electrochemical Structure of the Plasma–Liquid Interface." The Journal of Physical Chemistry C 126(2): 1222-1229.
Qian, M.-Y., C.-Y. Yang, Z.-d. Wang, X.-C. Chen, S.-Q. Liu and D.-Z. Wang (2016). "Numerical study of the effect of water content on OH production in a pulsed-dc atmospheric pressure helium–air plasma jet." Chinese Physics B 25(1).
Vichiansan, N., K. Leksakul, P. Chaopaisarn and D. Boonyawan (2021). "Simulation of simple 2D plasma jet model for NO, OH, and H2O2 production via Multiphysics in laminar flow and transport of diluted species through design of experiment method." AIP Advances 11(3).
Yan, W. and D. J. Economou (2017). "Gas flow rate dependence of the discharge characteristics of a helium atmospheric pressure plasma jet interacting with a substrate." Journal of Physics D: Applied Physics 50(41).
We only focused on determining potential accumulation site for RONS. Therefore, we have not considered the permeation of plasma feed gases into water. However, N2 permeation has been already investigated and given in the literature (J. Phys. Chem. A 2004, 108, 11573-11579) and OH is present in our manuscript. In addition, we have done an extensive literature review and unfortunately, the force field parameters of an atomic oxygen is not found. The force field parameters were developed mainly in ReaxFF which is used for reactive molecular dynamics simulations not for classical MD simulations.
